# Multi-Layer Fusion 3D Object Detection via Lidar Point Cloud and Camera Image

**Yuhao Guo and Hui Hu** *

College of Information Engineering, East China Jiaotong University, Nanchang 330013, China;
bigus_bond@163.com
* Correspondence: gnss523@163.com

**Abstract:** Object detection is a key task in automatic driving, and the poor performance of small object detection is a challenge that needs to be overcome. Previously, object detection networks could detect large-scale objects in ideal environments, but detecting small objects was very difficult. To address this problem, we propose a multi-layer fusion 3D object detection network. First, a dense fusion (D-fusion) method is proposed, which is different from the traditional fusion method. By fusing the feature maps of each layer, more semantic information of the fusion network can be preserved. Secondly, in order to preserve small objects at the feature map level, we designed a feature extractor with an adaptive fusion module (AFM), which reduces the impact of the background on small objects by weighting and fusing different feature layers. Finally, an attention mechanism was added to the feature extractor to accelerate the training efficiency and convergence speed of the network by suppressing information that is irrelevant to the task. The experimental results show that our proposed approach greatly improves the baseline and outperforms most state-of-the-art methods on KITTI object detection benchmarks.

**Keywords:** 3D object detection; multi-layer fusion; adaptive fusion; KITTI

---

## 1. Introduction

Object detection is a fundamental and crucial task in computer vision, which is aimed at identifying and locating specific classes of objects within the environment (e.g., cars, people). It has wide-ranging applications, particularly in the field of autonomous driving. Autonomous driving has requirements in many aspects: 3D detection, fast detection speeds, low costs, and so on. In response to the demand for 3D detection, cameras and Lidar have been installed in vehicles. With the development of deep learning, using neural network methods to detect objects has become a current trend. This paper focuses on object detection technology. High-performance object detection technology is of great significance in improving the safety of automatic driving. To achieve greater autonomous driving safety, more tasks and sensors will need to work together, and the technology in this paper is one of the many key technologies in autonomous driving systems.

At present, 3D object detection methods are divided into three kinds: image-based, point cloud-based and fusion-based. Images can provide vivid color and scale information and are the most commonly used data form in traditional object detection networks. Images have many advantages, such as containing continuous information, occupying small memory, having a low cost, and being easy to process. However, images are two-dimensional and cannot provide complete three-dimensional information. Lidar can provide accurate three-dimensional point cloud data, can directly obtain the spatial information of objects, can capture the details and shape of objects, and can replace the estimation of depth. It has relatively little impact on occlusion, viewing angle changes and lighting conditions and is suitable for accurate object detection in complex environments. However, the amount of Lidar data is large, the requirements for computing resources are high, and a large amount

of point cloud data needs to be processed and stored. In order to make comprehensive use of the advantages of these two kinds of data, the object detection method based on fusion has become a research hotspot in recent years. In this paper, an object detection network based on multi-sensor fusion is given priority, which is the inevitable direction of object detection in the field of autonomous driving.

Most fusion networks choose to fuse after extracting features from the image and point cloud data, respectively, which places higher requirements on the network; that is, it needs to retain more object information. With the deepening of the backbone network, the pixels of small objects in the feature map are decreased in number or even lost. Feature pyramid networks (FPN) [1] preserve small objects by fusing multi-scale features, but even simple additions can cause small objects' features to be overridden by large objects. Therefore, we believe that reasonable ways to guide the method of fusion are needed to preserve small object information. The reference to 'small objects' in this article refers specifically to situations in autonomous driving images and Lidar data that occupy fewer pixels or points in image or point cloud data due to the relative distance between the object and the sensor. Although these objects are small in size in terms of data, they are of great importance in ensuring road safety.

Aiming at the problem that the detection performance of small objects in automatic driving scenes is poor, affecting the detection accuracy of 3D objects, a multi-layer fusion 3D object detection network combining Lidar point cloud data and RGB images is proposed in this paper. This system is developed for use in autonomous vehicles to detect cars, pedestrians and cyclists. Compared with the method of using only RGB images, this method can use point cloud information to estimate the position and direction of objects more accurately, especially under poor lighting conditions. Compared with the method of using only Lidar, this method can make use of the texture and color information of the RGB images to achieve more accurate object recognition. In addition, in order to prevent the loss of small object information, either adding an adaptive fusion module (AFM) to the feature extraction network or using multi-layer fusion methods such as D-fusion in the fusion network has produced positive results. The main contributions of this paper are as follows:

(1)  A novel fusion method, D-fusion, is proposed, which can preserve the information of each layer of the fusion network to solve the problem of semantic loss and improve fusion performance.
(2)  We designed an adaptive fusion module (AFM) and applied it after using the feature extraction network, which effectively solves the problem of small-scale object loss in detection tasks.
(3)  An attention mechanism was introduced to optimize the efficiency of the feature extraction network.
(4)  We conducted comparative experiments on the challenging KITTI data set, and the results show that our network achieves satisfactory performance.

In this paper, Section 2 introduces related 3D object detection works, Section 3 introduces the overall framework of the network, Section 4 describes the experimental requirements, ablation experiments and analysis of the results, and Section 5 summarizes the work and looks at future directions.

## 2. Related Works

### 2.1. Image-Based 3D Object Detection

Image-based detection methods can be divided into monocular vision image and binocular vision image methods. Three-dimensional object detection networks based on monocular visual images mainly adopt the ideas of depth estimation [2,3], detecting key points [4] and using CAD prior information [5]. A monocular image is a two-dimensional projection of an eye cone in a three-dimensional world, with information such as depth missing. In order to obtain more accurate 3D information, such as depth features, some researchers are also studying 3D object detection networks based on binocular vision

images. Chen et al. proposed 3DOP [6] to estimate point clouds from binocular images. Xu and Chen proposed MLF [7] to estimate parallax maps from binocular images and reverse-project them into depth maps and point clouds. Li et al. proposed that CGStereo [8], combined with additional semantic segmentation supervision, significantly improves the accuracy of foreground depth estimation in images. Chen et al. proposed pseudo-stereo [9] to estimate depth maps from binocular images. Peng et al.'s study proposed to generate pseudo radar and target level depth estimation based on the SIDE [10] of two branch networks, respectively. It is difficult for image-based methods to obtain accurate 3D information, so the accuracy of detection is difficult to improve.

### 2.2. Point Cloud-Based 3D Object Detection

Point cloud-based object detection networks can be divided into three types. (1) A method based on the original point cloud. This method can retain the position information of objects in three-dimensional space to the maximum extent, such as 3DSSD [11]. (2) A method based on the projection of a point cloud. This method projects the point cloud into two-dimensional views from different angles and then uses a mature two-dimensional object detection network to achieve 3D object detection, such as RangeDet [12]. (3) A method based on point cloud voxelization. Disordered point cloud data are organized into ordered voxel expressive forms, and a 3D convolutional network is applied to extract the voxel features to achieve 3D object detection, such as SE-SSD [13]. Although this kind of network can accurately obtain the location information of the object, the information contained in the network is relatively sparse, resulting in heavy computation and loss of the object in the distance.

### 2.3. Fusion-Based 3D Object Detection

The method based on visual images can provide texture information but lacks depth information. The method based on point cloud provides spatial geometry information but lacks texture information. Texture information is helpful for object detection and classification, while depth information is helpful for object spatial location estimation. At present, it is a research direction of 3D object detection methods to improve the overall performance by using image and point cloud data at the same time. The methods based on fusion can be classified into three types: early fusion network, medium fusion network and late fusion network.

Early fusion refers to the fusion of information at the original pixel level. Pointpainting [14] and PI-RCNN [15] will color the point cloud by splicing the color information of each pixel of the image with the corresponding point cloud features and then use the existing detection network (such as PointRCNN [16], PointPillars [17], etc.) for the object detection of the colored point cloud. MVX-Net [18] uses two simple and effective early fusion methods, PointFusion and VoxelFusion, to integrate visual texture information and point cloud spatial geometry information to achieve high-precision object detection. This kind of network improves the effect well, but increases the amount of computation, and there is a problem that causes difficulty to align pixels. F-PointNet [19], proposed by Charles et al., Faraway-frustum [20], proposed by Zhang et al. and F-ConvNet [21], proposed by Wang et al., etc., generate high-quality two-dimensional candidate boxes using image data; they then map them to the three-dimensional space of the original point cloud. Three-dimensional candidate boxes were generated by extracting regional point cloud features. Unfortunately, such practices are largely limited by 2D detection results.

Medium fusion refers to the fusion of information at the feature map level or RoI (Regions of Interest) level. The MV3D [22] method proposed by Chen et al. uses the point cloud to generate the corresponding front view (FV) and bird's eye view (BEV), which, together with RGB images, serve as the input of the three feature extraction networks, respectively, and realizes the task of 3D object classifications and boundary box regression through deep fusion. Different from the MV3D, AVOD [23], proposed by Ku et al., only uses the BEV generated by the image data and point cloud data coding as the network

input and realizes the 3D object detection and classification and boundary box regression tasks through early fusion. SCANet [24], proposed by Lu and Chen, aims to effectively integrate multi-scale and global context information and, at the same time, generate attention from space and channels to select discriminant features. ContFuse [25] projects the image in a BEV to supplement the sparse BEV information. Crossfusion [26] realizes the cross-projection and fusion of an image and BEV on the basis of ContFuse. The cross-modality 3D object detection model [27] proposed by Zhu et al. not only realizes interactive fusion at the feature level but also combines 2D and 3D candidate boxes to optimize the results. RoIFusion [28], proposed by Chen et al., saves a lot of computation by integrating 3D RoI with 2D RoI. Medium fusion will not be very computation-intensive. However, in the process of fusion, the problem of information degradation still exists because of insufficient integration.

Late fusion refers to the fusion optimization of the detection results. A typical example is the CLOCs [29] proposed by Pang et al. The network first obtains 2D detection results and 3D detection results from the image and point cloud, respectively, and then filters the final 3D frame and adjusts the scale according to the geometric and semantic consistency of the 2D and 3D frames. This kind of network is difficult to apply because of the difficulty of training and real-time problems, so there are few related follow-up studies.

Based on the problem, the medium fusion network is not sufficient in the fusion stage. In this paper, the medium fusion network structure is adopted to fuse the point cloud and image at the ROI level, and the D-fusion method is designed to reduce semantic information degradation in the fusion network.

### 2.4. Detection of Small Objects

Small objects occupy very few pixels in the original image and will become very small in the feature map after convolution, which puts forward higher performance requirements on the network. In the current research, the method of feature fusion is basically adopted; that is, the shallow feature map and the deep feature map are fused together. Feature pyramid networks for object detection [1] are a typical example of this approach, which uses a pyramid structure to integrate the features of different layers. Dssd [30] deconvolves the deep feature map with the original dot product for feature fusion. Small object detection using context and attention [31] combines context information with an attention mechanism to comprehensively determine the object's category and location by understanding the object's background and paying attention to useful information. The main idea of augmentation for small object detection [32] is to over-sample small object samples so as to improve the performance of small object detection. In this paper, we use an attention mechanism to learn the information of the foreground and background and weigh the feature layers of different scales to reduce the loss of small object information.

### 3. The Proposed Approach

In this section, we describe the structure and implementation of a multi-layer fusion 3D object detection network. The proposed network architecture is shown in Figure 1. The network architecture consists of three main parts: feature extraction network, regional suggestion network (RPN) and fusion network. Using RGB images and the bird's eye view (BEV) as inputs, the feature extraction network processes them to obtain the corresponding feature maps. In this paper, VGG [33] is used as a backbone network. In addition, 100 K anchors are preset as the initial input to the RPN in 3D space. The RPN network filters these anchor boxes to obtain the RoI. Then, the feature maps are combined with the RoIs generated by RPN, and the corresponding feature region is cut out and sent to the fusion network for the final parameter regression. In the fusion network, we adopt a novel multi-layer fusion method, D-fusion, which can effectively combine features from different perspectives and retain the semantic information of each layer of the network so as to achieve 3D bounding box regression. Finally, the fusion network outputs classification and regression results. Both the feature extraction and fusion methods of the network adopt the

multi-layer fusion approach to achieve richer feature expression and more efficient data processing.

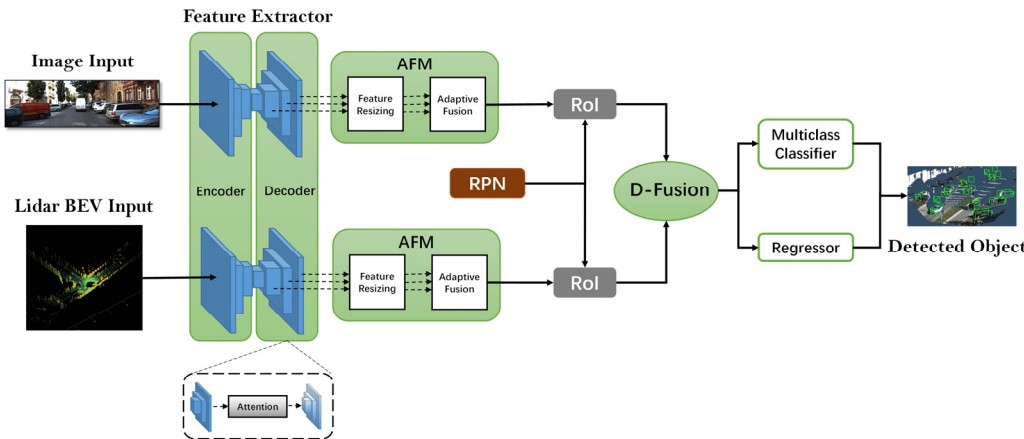

**Figure 1.** Overall frame of multi-layer fusion 3D object detection structure.

### 3.1. Inputs

The multi-layer fusion 3D object detection network has two types of input data: the RGB image and the Lidar point cloud. The camera is a typical representative of passive sensors. Images have rich color and texture information, which can help us intuitively understand the traffic scene and identify the object. Moreover, the image occupies very little memory. Therefore, we use the image as an input. However, this method lacks depth information, which is essential for accurate position estimation in the real 3D world. Using images as a standalone visual system is far from sufficient, as the brightness of the camera can easily degrade its accuracy at night or in rainy weather conditions. Lidar is a representative of active sensors. Lidar can not only acquire depth information but is less affected by external lighting conditions (i.e., at night) because it emits its own light pulses. Therefore, the Lidar system has higher accuracy and reliability than the camera system. We use the Lidar point cloud and RGB image input at the same time, which can complement their advantages and greatly improve the applicability of detection.

The point cloud data of the KITTI [34] dataset were collected using the Velodyne HDL-64E Lidar. Each collected point cloud file contains hundreds of thousands of points. Each point represents three-dimensional position and intensity information in a three-dimensional space and is distributed irregularly. Point cloud data are usually stored in the form of x, y, z and intensity, with the four values representing the 3D coordinates and reflectance of a point. Due to the uneven distribution of point cloud data and the large number of points, direct processing of point cloud data will take up a large amount of computing. Therefore, we use the BEV representation to represent the point cloud while preserving the information of the point cloud. Since all objects are not covered in the vertical direction of the road, the size and shape information of the object will be retained. The BEV includes the height map and density map of the point cloud obtained by encoding the height information and density information of the point cloud. The height map discretized the point cloud according to a certain resolution, projected the voxel onto the ground plane to generate a BEV, and took the highest height of the point in each voxel as the height feature. The slices are evenly divided over a certain height range so that the BEV contains more height features. The height feature map was calculated within each slice. The density map represents the number of points in each voxel. Considering the camera's viewing range, we selected the point cloud range of $[-40,40] \times [0,70]$ meters.

We discretized the projected point cloud into a two-dimensional grid with a resolution of 0.1 m. For each grid, the height feature is calculated as the maximum height of a point in the cell. To encode more detailed height information, the point cloud is evenly divided into five slices. By calculating a height plot for each slice, we obtain five distinct height

features. Point cloud density represents the number of points in each cell. To standardize the feature, it is calculated as $min(1.0, log(N + 1)log(64))$, where $N$ is the number of points in the cell. Note that the density feature is calculated for the entire point cloud, while the height feature is calculated for five slices, so the BEV is coded as a six-channel feature.

### 3.2. Feature Extractor

The multi-layer fusion 3D object detection network comprises two feature extraction networks, each dedicated to processing either the image or BEV input data. The structure of both feature extraction networks is the same. We designed the adaptive fusion module (AFM) and combined it with the attention mechanism to design the overall structure of the feature extractor. The network structure is shown in Figure 2.

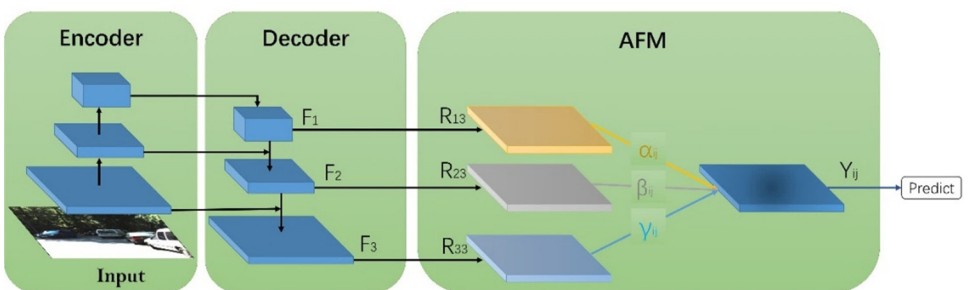

**Figure 2.** Feature extractor structure.

**Adaptive Fusion Module.** The features of each layer of the decoder are fused in the spatial dimension. Different from the previous methods of integrating multi-layer features using elements-wise sum or concatenation, the key idea of this method is to learn the spatial weights of the feature map fusion on each scale adaptively. It consists of two steps: feature map uniformity and adaptive fusion.

Feature map uniformity: As shown in Figure 2, feature maps with three different resolutions are represented as $F_1$, $F_2$ and $F_3$. We need to pre-process three feature maps with different resolutions into feature maps with the same dimensions. Since the dimension of $F_3$ was directly used in the final prediction, we adopted different upsampling strategies for the $F_1$ and $F_2$ feature maps, respectively, so that each dimension of $F_1$ and $F_2$ was consistent with that of $F_3$. For $F_1$ upsampling, we first use a $1 \times 1$ convolution layer to compress the number of channels of the feature to the number of channels of $F_3$, and then raise the resolution to be consistent with $F_3$ by nearest neighbor interpolation.

Adaptive fusion: we define $R_{13}$ and $R_{23}$, respectively, to represent the feature maps obtained after the consistency of the feature maps for $F_1$ and $F_2$, and $R_{33}$ and $F_3$ are the same feature. $\alpha(i, j)$, $\beta(i, j)$ and $\gamma(i, j)$ represent the weights of the $(i, j)$ vectors corresponding to $R_{13}$, $R_{23}$ and $R_{33}$, respectively. Note that $\alpha(i, j)$, $\beta(i, j)$ and $\gamma(i, j)$ are shared across all channels. Then, the fused feature can be expressed as:

$$Y(i, j) = \alpha(i, j) \times R_{13} + \beta(i, j) \times R_{23} + \gamma(i, j) \times R_{33} \tag{1}$$

The weight parameters $\alpha_{ij}$, $\beta_{ij}$ and $\gamma_{ij}$ are obtained by a $1 \times 1$ convolution of the three feature maps after uniformization. And the parameters $\alpha_{ij}$, $\beta_{ij}$ and $\gamma_{ij}$ are concatenated through the softmax function so that their range is in $[0, 1]$ and the sum is 1. Therefore, $\alpha(i, j)$, $\beta(i, j)$ and $\gamma(i, j)$ are obtained by the following calculation:

$$\begin{cases} \alpha(i, j) = \dfrac{e^{\alpha_{ij}}}{e^{\alpha_{ij}} + e^{\beta_{ij}} + e^{\gamma_{ij}}} \\ \beta(i, j) = \dfrac{e^{\beta_{ij}}}{e^{\alpha_{ij}} + e^{\beta_{ij}} + e^{\gamma_{ij}}} \\ \gamma(i, j) = \dfrac{e^{\gamma_{ij}}}{e^{\alpha_{ij}} + e^{\beta_{ij}} + e^{\gamma_{ij}}} \end{cases} \tag{2}$$

In this way, we obtain the $Y(i, j)$ for the image and point cloud; they are combined with the RPN network by the fusion network and fulfill classification and regression.

**Attention Mechanism.** Through the attention mechanism, the network can learn to selectively emphasize the informative features using global information and suppress the less useful features. In this network, we adopt SENet [35], a type of channel attention mechanism designed to enhance the network's representation ability by enabling it to perform dynamic channel feature recalibration.

We choose to add the SENet module to the decoder of the feature extraction network. The SENet module consists of two operations, squeeze and excitation, and is composed of a pooling layer, a convolutional layer and an activation layer. As shown in Figure 3, the original feature map $X$ is first globally average-pooled to obtain $S$, with the dimension changing from $H \times W \times C$ to $1 \times 1 \times C$, corresponding to the squeeze operation. Then, $S$ is processed by the convolutional layer and activation layer to obtain the weighted information $E$, corresponding to the excitation operation. Finally, $E$ is multiplied by the original feature map $X$ on a channel-wise basis to obtain the final $X^*$. The purpose of the squeeze operation is to enhance the correlation of channel data. The purpose of the excitation operation is to obtain the weight coefficients for each feature map on the channel dimension, thus making the channel features of the feature map more capable of extracting features, amplifying effective features, and reducing ineffective feature information. In short, the channel attention mechanism SENet is designed to allow the network to use more effective channels and suppress relatively ineffective ones.

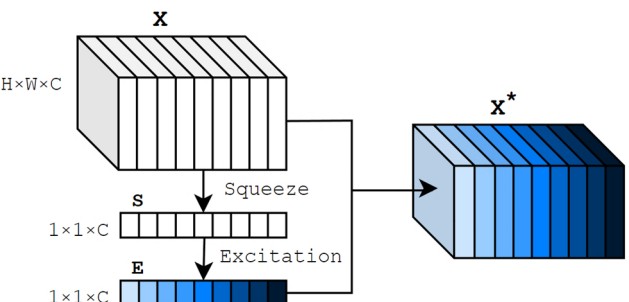

**Figure 3.** Structure of attention mechanism SENet.

### 3.3. RPN

We have adopted the same RPN network as the AVOD network and do not claim novelty here.

The representation of each anchor box is defined by six parameters, namely $(c_x, c_y, c_z, l, w, h)$, where $(c_x, c_y, c_z)$ represents the centroid coordinates of the anchor box, and $(l, w, h)$ specifies the dimensions of the box. In this work, we leverage the benefits of BEV images to generate anchor boxes that are invariant to occlusion and preserve the object sizes. Specifically, we sample the anchor boxes at 0.5 m intervals along the BEV plane, with $(c_x, c_y)$ serving as the center. The vertical coordinate $c_z$ is determined by the height of the Lidar sensor from the ground. We use the K-means clustering method to cluster the labels in the training set to determine the initial size of the anchor. Due to the sparsity of the BEV, many anchor boxes may not contain any point clouds. To eliminate such empty anchor boxes, we utilize an integral image to calculate the point occupancy map.

The anchor boxes are projected onto two feature maps obtained from the BEV and RGB images, resulting in a $7 \times 7$ feature crop for each box. These crops are down-sampled via a $1 \times 1$ convolution kernel to reduce the number of parameters in subsequent operations. The resulting feature crops undergo the element-wise mean operation and are then input to a fully connected block that outputs the region proposal parameters, including the object's confidence and offset. A 2D non-maximum suppression (NMS) algorithm is applied to remove overlapping proposals and retain up to a maximum of 1024 proposals. The fully

connected block consists of three fully connected layers with a size of 2048, which output the bounding box regression, direction estimation and object classification.

### 3.4. D-Fusion

We designed a fusion approach—dense fusion (D-fusion). Compared with the previous early fusion, late fusion and deep fusion, it can not only combine the features from multiple views but also effectively combine the semantic information of each layer in the network to carry out three-dimensional box regression. The network structure is shown in Figure 4d.

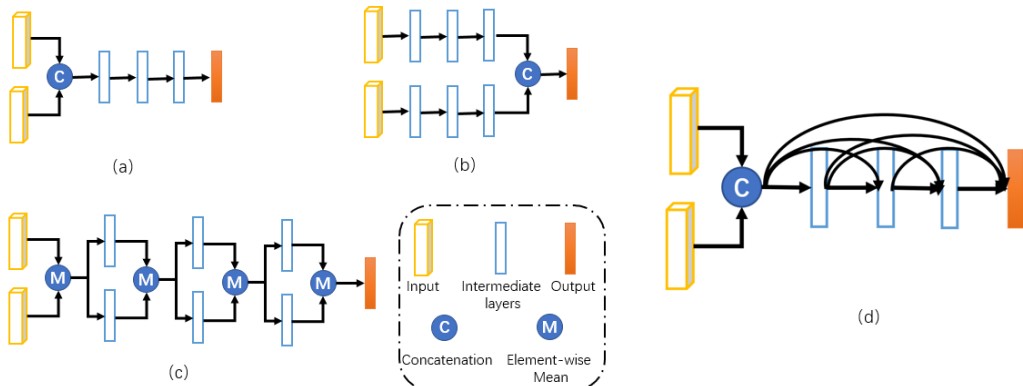

**Figure 4.** Different types of fusion: (**a**) represents the early fusion method, (**b**) represents the late fusion method, (**c**) represents the deep fusion method, and (**d**) represents the dense fusion (D-fusion) method.

Since features of different views often have different resolutions, we use RoI pooling for each view to obtain feature vectors of the same length. For the generated 3D suggestions, we are able to project them into any view in the 3D space. In our example, we project them onto two views, the BEV and the RGB Image.

In order to combine information on different features, the previous work usually adopted early fusion, late fusion or deep fusion. Inspired by DenseNet, we adopted the dense fusion method to fuse every layer of the network densely. A comparison of the architecture of our D-fusion network and the early/late/deep fusion network is shown in Figure 4. For networks with P-layers, early fusion combines $F_{BEV}$ and $F_{RGB}$ from multiple views at the input stage:

$$F_P = H_P(H_{P-1}(\cdots H_1(F_{BEV} \oplus F_{RGB}))) \tag{3}$$

$\{H_p,\ p = 1,\ldots,P\}$ is the feature transformation function. $H_p$ refers to the feature map obtained at the P-layer. $\oplus$ is a join operation (such as concatenation and sum).

In contrast, late fusion uses separate subnetworks to independently learn feature transforms and combine their outputs in the prediction stage:

$$F_P = (H_P(H_{P-1}(\cdots H_1(F_{BEV})))) \oplus (H_P(H_{P-1}(\cdots H_1(F_{RGB})))) \tag{4}$$

In early and late fusion, the operation $\oplus$ is implemented using the concatenation method. Deep fusion enables more interaction between the middle features from different aspects:

$$F_0 = F_{BEV} \oplus F_{RGB}$$
$$F_P = \left(H_P^{BEV}(F_{P-1}) \oplus \left(H_P^{RGB}(F_{P-1}), \forall p = 1, 2, \cdots P \right.\right. \tag{5}$$

The operation $\oplus$ in deep fusion uses the method of element-wise mean.

To further improve the flow of information between layers, we propose a different fusion mode. It works by connecting directly from any layer to all subsequent layers:

$$
\begin{aligned}
F_0 &= F_{BEV} \oplus F_{RGB} \\
F_1 &= H_1(F_{BEV} \oplus F_{RGB}) \\
F_P &= a_{p-1}\big(a_{p-2}(\cdots a_2(a_1 F_1 \oplus) \cdots \oplus F_{P-2}) \oplus F_{P-1}\big), \\
&\quad \forall p = 2, 3, \cdots, P
\end{aligned}
\tag{6}
$$

The operation $\oplus$ in the D-fusion uses the weighted summation method between concatenation and element-wise mean. Where $a_1, \ldots \ldots a_{p-2}, a_{p-1}$, respectively, represent the weights given after the feature fusion of each layer. The network adopts a three-layer network structure with the default settings $a_1 = 1/2$, $a_2 = 1/2$ and $a_3 = 1$. We also use the dropout mechanism to mitigate the occurrence of overfitting and achieve a regularization effect to some extent. It also saves computing overhead.

### 3.5. Training

We trained two separate networks, one for the car class and the other for the pedestrian and cyclist classes. The RPN and detection networks were jointly trained in an end-to-end approach using mini-batches that contained one image with 512 and 1024 RoIs, respectively. The ADAM optimizer was used with an initial learning rate of 0.0001, which decayed exponentially every 30 K iterations with a decay factor of 0.8. The network was trained for 120 K iterations.

### 4. Experiment and Results

We tested the performance of the multi-layer fusion 3D object detection network for proposal generation and object detection tasks on three classes of the KITTI object detection benchmark. According to the 7481 training frames provided by the KITTI dataset, we divided the training set and testing set into a ratio of about 1:1. For the evaluation, we followed KITTI's easy, moderate and hard difficulty levels. We evaluated and compared the four versions we implemented as follows: the first version using early fusion, the second version using late fusion, the third version using deep fusion and the fourth version using our D-fusion.

The training and testing of the network was run on an NVIDIA GeForce GTX 1080 Ti GPU (NVDIA, Santa Clara, CA, USA) with 11GB of memory. This network includes feature extraction networks based on AFM and SENet, as well as fusion networks based on D-fusion. Figure 5 shows the final output result. The comparison of small object detection results is shown in Figure 6. We only take the car class for demonstration because the small object problem and occlusion are more common and prominent in the car class. As can be seen in Figure 6, both small-size objects and occluded small-size objects can be effectively detected.

The training results of detection accuracy are shown in Figure 7. As the number of iterations increases, the accuracy of object detection continues to improve. We use three indicators to evaluate the performance of the network, namely $AP_{2D}$, $AP_{3D}$ and $AP_{BEV}$. Figure 7 shows the performance of the network on the car class. The performance of the network for pedestrian and cyclist classes is shown in Figure 7. As the number of training epochs increases, the accuracy also continues to improve. For cars, the accuracy of a 2D prediction box is close to 90%, while the accuracy of a 3D prediction box is about 85%. The accuracy of 2D and 3D prediction boxes on the pedestrian class is close to 60%. The accuracy of the cyclist class is close to 65%. As shown in Figure 7, the detection accuracy of cars is much higher than that of pedestrians and cyclists. This is because the sample number of cars is relatively large and abundant.

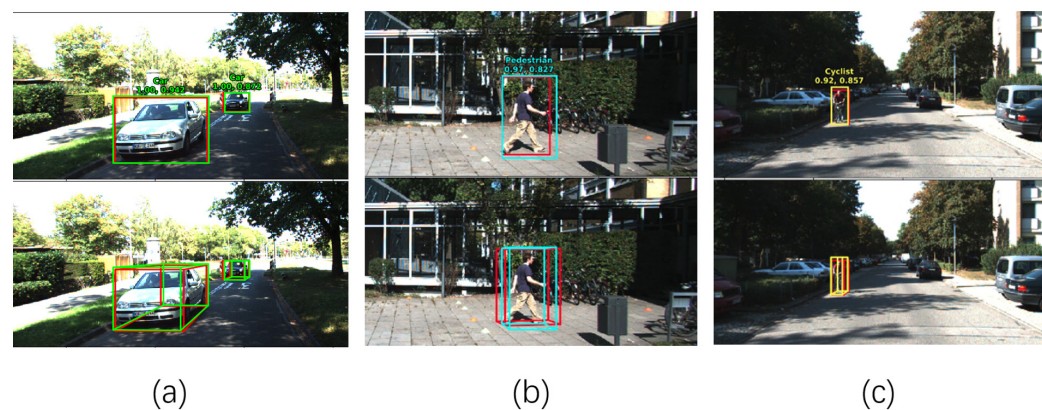

(a)          (b)          (c)

**Figure 5.** Test result diagram. Car, pedestrian and cyclist represent three object classes, with the predicted car represented by a green box in (**a**), pedestrian represented by a turquoise box in (**b**), and cyclist represented by a yellow box in (**c**). The red boxes represent the true value of GroundTruth. The first score represents the confidence score of the 3D prediction box, and the second score represents the intersection over union (IoU) of the 3D prediction box and the GroundTruth.

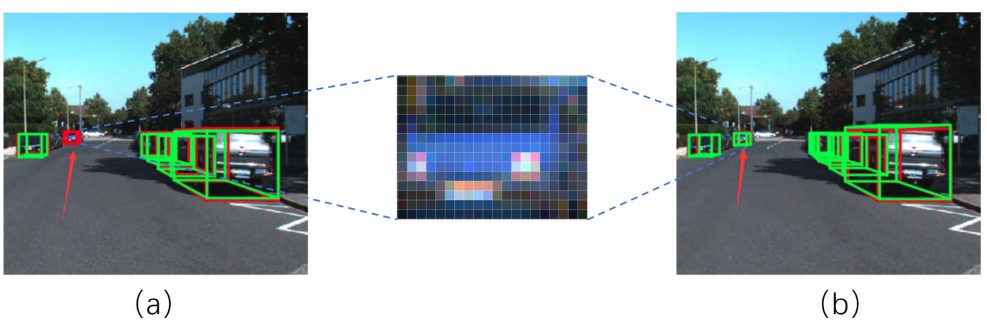

(a)          (b)

**Figure 6.** Comparison of small object detection results. (**a**) represents the network AVOD, and (**b**) represents the network in this paper. The red boxes represent the GroundTruth. The green boxes represent the predicted results.

### 4.1. 3D Detection

For the final 3D detection results, we used two metrics to measure the accuracy of 3D positioning and 3D anchor box detection. For 3D positioning, we projected the 3D box to the ground plane to obtain the BEV anchor frame. We calculated the average accuracy of the BEV anchor frame ($AP_{BEV}$). For the 3D bounding box, we used the average precision ($AP_{3D}$) metric to evaluate the complete 3D anchor box.

When using $AP_{3D}$ and $AP_{BEV}$ for evaluation, we set an IoU threshold of 0.7 for the car class and 0.5 for the pedestrian and bicycle classes. We compared the detection results with the state-of-the-art network publicly available on the validation set. On the validation set, as shown in Table 1, our architecture performs optimally in both car and pedestrian detection. It is worth mentioning that in the comparison of car and pedestrian detection, our architecture is, on average, 3.19% and 5.55% higher than AVOD-FPN on $AP_{3D}$ and the $AP_{BEV}$, respectively.

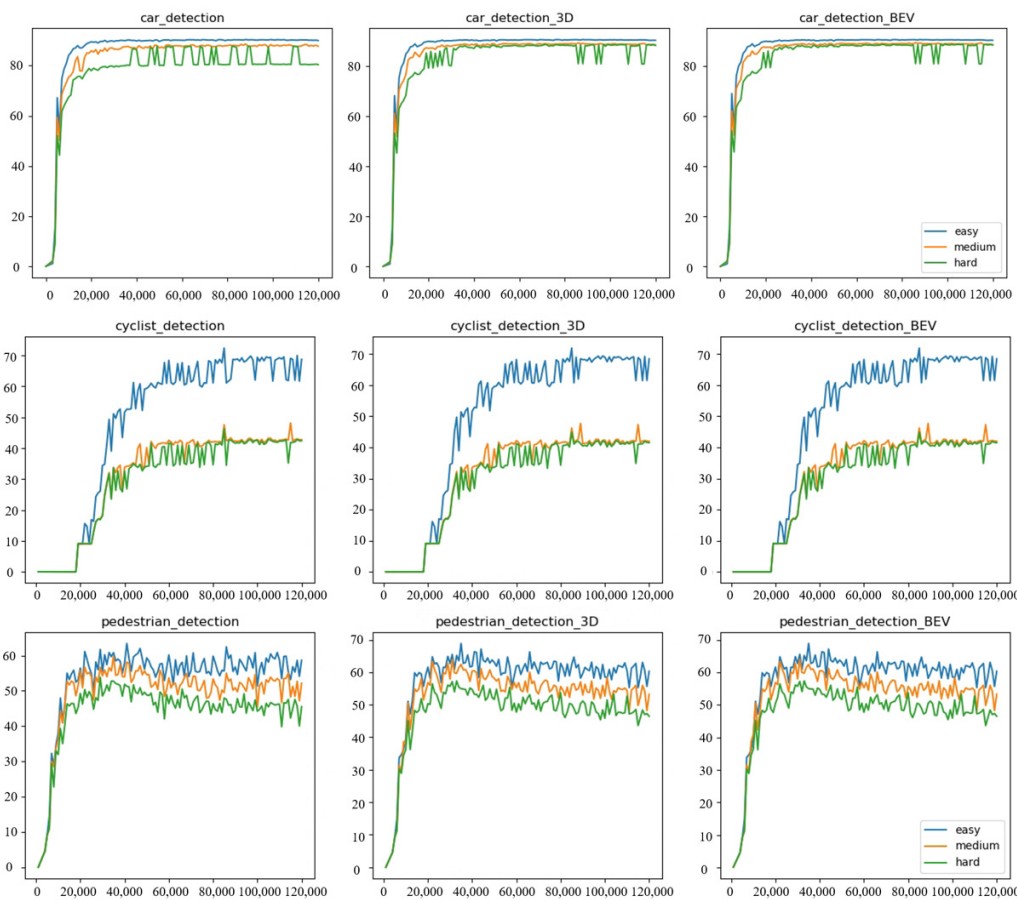

**Figure 7.** The testing accuracy of the object detection. The following three indicators measure the test results of the car, cyclist and pedestrian. Where object_detection, object_detection_3D and car_detection_BEV represent the AP, where an object represents the car, cyclist or pedestrian.

**Table 1.** Average precision of 3D anchor boxes on the KITTI validation set ($AP_{3D}$) (represented by %) and average precision of BEV anchor boxes ($AP_{BEV}$) (represented by %). The best scores are highlighted in bold.

| Method | Class | $AP_{3D}$ (%) | | | $AP_{BEV}$ (%) | | |
|---|---|---|---|---|---|---|---|
| | | Easy | Moderate | Hard | Easy | Moderate | Hard |
| MV3D | | 71.09 | 62.35 | 55.12 | 86.02 | 76.90 | 68.49 |
| AVOD | | 73.59 | 65.78 | 58.38 | 86.80 | 85.44 | 77.73 |
| AVOD-FPN | | 81.94 | 71.88 | 66.38 | 88.53 | 83.79 | 77.90 |
| F-PointNet | | 81.20 | 70.39 | 62.19 | 88.70 | 84.00 | 75.33 |
| SCANet | Car | 83.63 | 74.47 | 67.78 | - | - | - |
| MVX-Net | | 83.20 | 72.70 | 65.20 | 89.20 | 85.90 | 78.10 |
| ContFuse | | 82.54 | 66.22 | 64.04 | 88.81 | 85.83 | 77.33 |
| CrossFusion | | 83.20 | 74.50 | 67.01 | 88.39 | 86.17 | 78.23 |
| Ours | | **85.58** | **75.37** | **68.83** | **89.59** | **86.62** | **79.52** |
| MV3D | | 39.48 | 33.69 | 31.51 | 46.13 | 40.74 | 38.11 |
| AVOD | | 38.28 | 31.51 | 26.98 | 42.52 | 35.24 | 33.97 |
| AVOD-FPN | Ped | 50.80 | 42.81 | 40.88 | 58.75 | 51.05 | 47.54 |
| F-PointNet | | 51.21 | 44.89 | 40.23 | 58.09 | 50.22 | 47.20 |
| Ours | | **53.87** | **51.27** | **45.99** | **62.00** | **56.19** | **49.97** |

**Table 1.** *Cont.*

| Method | Class | AP$_{3D}$ (%) | | | AP$_{BEV}$ (%) | | |
|---|---|---|---|---|---|---|---|
| | | **Easy** | **Moderate** | **Hard** | **Easy** | **Moderate** | **Hard** |
| MV3D | | 61.22 | 48.36 | 44.37 | 66.70 | 54.76 | 50.55 |
| AVOD | | 60.11 | 44.90 | 38.80 | 63.66 | 47.74 | 46.55 |
| AVOD-FPN | Cyc | 64.00 | 52.18 | 46.61 | 68.09 | 57.48 | 50.77 |
| F-PointNet | | **71.96** | **56.77** | **50.39** | **75.38** | **61.96** | **54.68** |
| Ours | | 68.66 | 42.23 | 41.71 | 68.28 | 46.48 | 40.64 |

### 4.2. The Effect of D-Fusion

To analyze the effectiveness of different fusion methods, we tested four types of fusion methods. They are early fusion, late fusion, deep fusion and the D-fusion we designed. From Table 2, it can be clearly seen that in the case where only the fusion method is different, D-fusion basically exhibits optimal performance in detection tasks of different types of objects or different difficulty levels. Compared with early fusion, late fusion and deep fusion of AP$_{3D}$, the network with D-fusion increased by 8.35%, 8.33% and 9.75%, respectively. In comparison with AP$_{BEV}$, the network with D-fusion increased by 8.94%, 5.94% and 8.35%, respectively.

**Table 2.** Average precision comparison of different fusions on the KITTI validation set. The best scores are highlighted in bold.

| Method | Class | AP$_{3D}$ (%) | | | AP$_{BEV}$ (%) | | |
|---|---|---|---|---|---|---|---|
| | | **Easy** | **Moderate** | **Hard** | **Easy** | **Moderate** | **Hard** |
| Early fusion | | 82.12 | 73.87 | 67.70 | 88.50 | 86.07 | 79.03 |
| Late fusion | | 70.28 | 56.48 | 55.85 | 86.47 | 77.17 | 70.03 |
| Deep fusion | Car | 82.86 | 73.42 | 67.30 | 89.07 | 85.91 | 78.99 |
| D-fusion (Ours) | | **85.58** | **75.37** | **68.83** | **89.59** | **86.62** | **79.52** |
| Early fusion | | 45.60 | 40.75 | 35.07 | 48.71 | 43.42 | 37.23 |
| Late fusion | | 48.77 | 43.75 | 37.20 | 52.64 | 46.55 | 45.16 |
| Deep fusion | Ped | 47.16 | 40.85 | 35.21 | 54.43 | 47.69 | 41.83 |
| D-fusion (Ours) | | **53.87** | **51.27** | **45.99** | **62.00** | **56.19** | **49.97** |
| Early fusion | | 49.62 | 32.1 | 31.57 | 50.09 | 32.49 | 31.89 |
| Late fusion | | 65.19 | 40.71 | 40.32 | 65.72 | 41.35 | **40.73** |
| Deep fusion | Cyc | 46.22 | 29.07 | 23.66 | 47.16 | 29.66 | 29.37 |
| D-fusion (Ours) | | **68.66** | **42.23** | **41.71** | **68.28** | **46.48** | 40.64 |

### 4.3. The Effect of AFM and SENet

For the feature extractors, we compared three scenarios: traditional convolution with FPN, traditional convolution with AFM and traditional convolution with AFM and SENet, as shown in Table 3. For the car class, traditional convolution with AFM and SENet has a certain effect on AP$_{3D}$, but on the AP$_{BEV}$, the difference is very small compared to traditional convolution with AFM, indicating that the attention mechanism does not play a significant role in the BEV. For the pedestrian categories, traditional convolution with AFM and SENet is significantly better than traditional convolution with FPN, but it is also significantly worse than traditional convolution with AFM. We analyzed that the reason for this result is that there are many small object pedestrian samples, and factors such as human posture cause significant sample differences. For the cyclist class, the results of traditional convolution with AFM and SENet are significantly better than those of traditional convolution with FPN and also better than traditional convolution with AFM. It is worth noting that compared to traditional convolution with FPN, traditional convolution with AFM and SENet is about 20% higher at the easy level, at least 10% higher at the moderate level and about 16% higher at the hard level.

**Table 3.** Average precision comparison of different feature extractors on the KITTI validation set. The best scores are highlighted in bold.

| Method | Class | AP$_{3D}$ (%) | | | AP$_{BEV}$ (%) | | |
|---|---|---|---|---|---|---|---|
| | | Easy | Moderate | Hard | Easy | Moderate | Hard |
| FPN | | 83.25 | 74.55 | 67.46 | 89.24 | 86.57 | 78.81 |
| AFM | Car | 84.71 | 74.79 | 68.17 | **89.77** | **86.84** | 79.34 |
| AFM and SENet | | **85.12** | **75.65** | **68.78** | 89.74 | 86.66 | **79.39** |
| FPN | | 47.33 | 41.23 | 35.29 | 55.40 | 48.57 | 42.27 |
| AFM | Ped | **61.32** | **54.47** | **47.62** | **59.30** | **52.60** | **46.17** |
| AFM and SENet | | 50.69 | 45.17 | 39.88 | 57.01 | 50.37 | 44.09 |
| FPN | | 48.48 | 31.25 | 25.66 | 49.16 | 31.75 | 25.92 |
| AFM | Cyc | 66.65 | 40.97 | 40.46 | 67.21 | 40.80 | 40.51 |
| AFM and SENet | | **68.03** | **41.88** | **41.45** | **69.20** | **42.71** | **41.74** |

### 4.4. Ablation Experiment

In order to investigate the contribution of various improvement methods to the network, we conducted ablation experiments on three class detection tasks. From the data in Table 4, it can be seen that D-fusion, AFM and SENet have all improved network performances. Moreover, in the car detection tasks, the combination of the three can achieve the best overall results. In the detection task of pedestrians and cyclists, the three together failed to achieve the optimal effect on each task, which is due to a certain degree of overfitting caused by the deepening of the network. Pay attention to network complexity as well as performance improvement. Table 5 shows the number of parameters in each part of the network. Thus, the introduction of the SENet and AFM modules increases the number of parameters by a relatively small amount.

**Table 4.** Ablation study. The best scores are highlighted in bold.

| Method | Class | AP$_{3D}$ (%) | | | AP$_{BEV}$ (%) | | |
|---|---|---|---|---|---|---|---|
| | | Easy | Moderate | Hard | Easy | Moderate | Hard |
| D-fusion | | 83.99 | 74.93 | 68.08 | 89.41 | 80.12 | 79.21 |
| AFM | | 84.52 | 74.60 | 68.00 | 89.33 | 86.37 | 79.32 |
| SENet | | 82.90 | 73.40 | 66.98 | 88.59 | 85.73 | 78.65 |
| D-fusion and AFM | Car | 85.11 | **75.70** | 68.64 | 89.19 | 86.56 | 79.29 |
| D-fusion and SENet | | 84.45 | 74.80 | 67.64 | 89.53 | 80.13 | 79.05 |
| AFM and SENet | | 85.12 | 75.65 | 68.78 | **89.74** | 86.66 | 79.39 |
| D-fusion and AFM and SENet | | **85.58** | 75.37 | **68.83** | 89.59 | **86.72** | **79.52** |
| D-fusion | | 53.98 | 48.44 | 41.90 | 57.95 | 51.97 | 45.09 |
| AFM | | **59.27** | **52.56** | **46.14** | 61.35 | 54.54 | 47.64 |
| SENet | | 47.33 | 41.23 | 35.29 | 55.40 | 48.57 | 42.27 |
| D-fusion and AFM | Ped | 53.98 | 48.78 | 42.87 | 57.13 | 51.41 | 45.09 |
| D-fusion and SENet | | 53.84 | 50.65 | 45.14 | 58.79 | 53.42 | 47.65 |
| AFM and SENet | | 54.02 | 43.79 | 41.83 | 54.83 | 49.23 | 43.23 |
| D-fusion and AFM and SENet | | 53.87 | 51.27 | 45.99 | **62.00** | **56.19** | **49.97** |
| D-fusion | | 68.07 | 41.84 | 41.22 | 58.61 | 40.46 | 39.90 |
| AFM | | 66.65 | 40.97 | 40.46 | 67.21 | 40.80 | 40.51 |
| SENet | | 48.48 | 31.25 | 25.66 | 49.16 | 31.75 | 25.92 |
| D-fusion and AFM | Cyc | **69.11** | **43.42** | **42.40** | **69.72** | 43.95 | **42.95** |
| D-fusion and SENet | | 61.19 | 41.38 | 34.57 | 61.70 | 41.67 | 34.81 |
| AFM and SENet | | 67.93 | 42.11 | 41.07 | 68.53 | 42.31 | 41.92 |
| D-fusion and AFM and SENet | | 68.66 | 42.23 | 41.71 | 68.28 | **46.48** | 40.64 |

**Table 5.** Network parameters.

| Architecture | Number of Parameters |
| --- | --- |
| Base Model | 26,265,899 |
| Backbone (Image and Lidar) | 9,366,336 & 9,366,336 |
| AFM | 64,515 |
| SENet | 717,440 |
| D-fusion | 12,589,056 |
| Total | 38,854,955 |

## 5. Conclusions

This paper presents a 3D object detection network that leverages LiDAR point clouds and RGB images, with its effectiveness validated through experiments on the KITTI dataset. Firstly, we propose a new D-fusion method based on the existing three fusion methods, which solves the problem of semantic loss in the fusion network. Secondly, we have improved the feature extraction network by adding AFM and attention mechanism to the traditional convolutional network, which improves the detection accuracy. At the same time, the network also performs well in small object detection tasks. By comparing with existing fusion networks of the same type, our network achieved the overall best performance on the KITTI benchmark.

According to the analysis of the poor experimental results, a large portion of detection errors is attributed to the similarity between the background and the object in a complex environment. Afterward, we will introduce data augmentation and instance segmentation to enhance the network's ability to cope with complex environments. In addition, the running speed of the network has not yet met the requirements for processing video stream data. We will improve the detection speed by streamlining the network structure.

**Author Contributions:** Conceptualization, Y.G. and H.H.; methodology, Y.G.; software, Y.G.; validation, Y.G.; formal analysis, Y.G.; investigation, H.H.; resources, H.H.; data curation, Y.G.; writing—original draft preparation, Y.G.; writing—review and editing, H.H.; visualization, Y.G.; supervision, H.H.; project administration, Y.G.; funding acquisition, H.H. All authors have read and agreed to the published version of the manuscript.

**Funding:** This study was supported by the National Natural Science Foundation of China under Grant No. 61961020.

**Institutional Review Board Statement:** Not applicable.

**Informed Consent Statement:** Not applicable.

**Data Availability Statement:** The code is available at https://github.com/JackKu0/MLFOD (accessed on 9 August 2023). We used the open-source data KITTI, which is available at https://www.cvlibs.net/datasets/kitti/eval_object.php?obj_benchmark=3d (accessed on 10 August 2019).

**Conflicts of Interest:** The authors declare no conflict of interest.

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
