# Peer review of "Multi-Layer Fusion 3D Object Detection via Lidar Point Cloud and Camera Image"

_applsci, doi:10.3390/app14041348_

Round 1

Reviewer 1 Report

Comments and Suggestions for Authors

The language of this manuscript is fine.

It is very difficult to keep the related work section of such work up-to-date and relevant, since this topic has been and is being studied so much. I can hardly maintain to be aware of everything myself on it. The classic approach to object detection (recognizing 3D objects from their 2D appearances) used to investigate the invariances needed. This is task-dependent. In the traffic scenes shift invariance is accounted for by the convolutional structure of the networks, rotational invariance in the image plane is not required (though cyclists or pedestrians lying horizontally should be an issue!), rotational invariance on the scene plane must be accounted for by learning many appearances, and the unknown distance of the objects requires scale-invariance. The latter is very limited in classical CNN object recognition. It must either also be learned, or taken care of by special changes to the architecture which are the topic of the paper. The topic of invariance is not discussed anymore, and accordingly the word does also not appear in this manuscript. A pity.

The term 'small object' is ubiquitously used throughout the text, a core term. Note, there is a whole branch of pattern recognition devoted to the topic of small targets, including own benchmark data sets (e.g. the SIRST). In this paper 'small object' does not refer to 3D-small objects - such as birds, cats, dogs or kids - which may well appear in such scenery and should not go unnoticed, though they might not be labeled and present in KITTI. Instead, here it refers to objects appearing small - because of their greater distance. Thus they occupy only few cells as well in the layers of the visual channel as in the layers of the LIDAR channel. Due to the general low-pass character of the filters they tend to get lost - which is here countered by various attention mechanisms and bridges in the architecture.

Near line 29, right in the beginning, the requirements for autonomous driving are listed. The most important one is missing: SAFETY. It will never be accepted neither by the authorities nor the public, if it is not proven to be at least as safe as human guided driving. And with respect to this, the numbers given in the results section in the tables are far from sufficient, in particular about the cyclists and pedestrians. Let us hope that the real systems, on which the industry works now for quite some decades, are trained on far greater labeled data-sets than just the poor small old KITTI. Much more invariance with respect to lighting, weather, season, geographic region etc. are required, still leaving the rare exceptional event issue open. But hopefully, they should achieve much higher recognition performances, than the ones reported here. Academia remains restricted to the rather limited public benchmarks. We lack behind. It is all a rather futile effort. 

The original part of the paper is the D-Fusion part, which addresses the information-in-few-cells-only problem. This is nicely explained in Figure 4 and the corresponding text. A more complex architecture with more bridges over the scales is introduced. According to the ablation study in Chapter 4, this yields some progress, and according to the samples given in the Figures this gain is won on the remote targets, appearing small in the data. So the publication of this idea is justified, and the paper should be published.

However, I would not let such work pass, until the additional complexity introduced is quantified. It is good custom these days to publish the numbers of parameters to be trained also in the results tables or in a separate table. How may parameters does the attention mechanism introduce? How many are needed for the fusion variants? How many are needed already for the visual and LIDAR backbone, respectively? Here is a serious trade-off, which is only very lightly discussed in the paper (where the 1x1 convolution is mentioned). By the way, I know everybody speaks of 1x1 convolution today, but that really distorts the meaning of the nice old engineer-term 'convolution' into absurdness. The old pattern recognition rule of thump had it that you need ten samples to properly train one parameter. That is in accordance with Vapnic Chervonenkis theory. How many million parameters do we have here, and how many are added by the original part of the paper, and how many labeled samples does KITTI provide?

Very serious issues on which I would like to see some awareness in a revision of this work! 

Reviewer 2 Report

Comments and Suggestions for Authors

This manuscript explores the intriguing subject of 3D object recognition using the fusion photos and point clouds. The related work section necessitates augmentation as it now lacks adequacy in addressing the topic matter. Furthermore, the outcome does not correspond to the given title. The data are presented in a two-dimensional format (Figure 6) whereas the viewer anticipates three-dimensional recognition. The description of the point cloud input is insufficient. A more comprehensive description of the complete network structure is required. It would be quite beneficial to provide a comparison with another well-established method, such as the "Complexer-YOLO: Real-Time 3D Object Detection and Tracking on Semantic Point Cloud". Implementing the aforementioned modifications would result in the writing of a manuscript that meets the desired standards of this journal.

Comments on the Quality of English Language

It is advisable to give it for professional editing.

Round 2

Reviewer 1 Report

Comments and Suggestions for Authors

My points have been addressed properly. In particular, I appreciate the inclusion of the table with the numbers of parameters.

Reviewer 2 Report

Comments and Suggestions for Authors

Satisfied with the revision